# The human SKA complex drives the metaphase-anaphase cell cycle transition by recruiting protein phosphatase 1 to kinetochores

Sushama Sivakumar[1,2,3], Paweł Ł Janczyk[4], Qianhui Qu[2,3], Chad A Brautigam[5], P Todd Stukenberg[4]*, Hongtao Yu[2,3]*, Gary J Gorbsky[1]*

[1]Cell Cycle and Cancer Biology Research Program, Oklahoma Medical Research Foundation, Oklahoma City, United States; [2]Department of Pharmacology, University of Texas Southwestern Medical center, Dallas, United States; [3]Howard Hughes Medical Institute, University of Texas Southwestern Medical center, Dallas, United States; [4]Department of Biochemistry and Molecular Genetics, University of Virginia School of Medicine, Charlottesville, United States; [5]Department of Biophysics, University of Texas Southwestern Medical center, Dallas, United States

*For correspondence: pts7h@virginia.edu (PTS); hongtao.yu@utsouthwestern.edu (HY); GJG@omrf.org (GJG)

**Competing interests:** The authors declare that no competing interests exist.

**Abstract** The spindle- and kinetochore-associated (Ska) complex is essential for normal anaphase onset in mitosis. The C-terminal domain (CTD) of Ska1 binds microtubules and was proposed to facilitate kinetochore movement on depolymerizing spindle microtubules. Here, we show that Ska complex recruits protein phosphatase 1 (PP1) to kinetochores. This recruitment requires the Ska1 CTD, which binds PP1 in vitro and in human HeLa cells. Ska1 lacking its CTD fused to a PP1-binding peptide or fused directly to PP1 rescues mitotic defects caused by Ska1 depletion. Ska1 fusion to catalytically dead PP1 mutant does not rescue and shows dominant negative effects. Thus, the Ska complex, specifically the Ska1 CTD, recruits PP1 to kinetochores to oppose spindle checkpoint signaling kinases and promote anaphase onset. Microtubule binding by Ska, rather than acting in force production for chromosome movement, may instead serve to promote PP1 recruitment to kinetochores fully attached to spindle microtubules at metaphase.

## Introduction

Anaphase onset and mitotic exit are driven by proteasome-mediated destruction of Securin and Cyclin B, which are targeted for ubiquitylation by the E3 ligase, the Anaphase-promoting complex/cyclosome (APC/C) (*Sivakumar and Gorbsky, 2015*). Until metaphase, APC/C activity is restrained by the spindle checkpoint. Recent studies indicate that a key element in extinguishing checkpoint signaling when chromosomes align at metaphase is the displacement of the critical checkpoint signaling kinase, Mps1, from its substrate, Knl1, at kinetochores (*Aravamudhan et al., 2015*; *Hiruma et al., 2015*; *Ji et al., 2015*). Then, to allow anaphase onset and mitotic exit, Mps1 phosphorylation events must be reversed by phosphatases. Previously, we showed that depletion of the Ska complex leads to a strong metaphase arrest or delay (*Daum et al., 2009*; *Sivakumar et al., 2014*). Even when Mps1 was directly inhibited with a strong chemical inhibitor, cells depleted of the Ska complex exited mitosis more slowly (*Sivakumar et al., 2014*). These results suggested that the Ska complex played some role in opposition to, or downstream of, checkpoint signaling. Here, we provide evidence that the Ska complex recruits protein phosphatase 1 (PP1) to kinetochores, consistent with a role in opposing checkpoint signaling by kinetochore-associated kinases.

**eLife digest** When one cell divides into two daughter cells it is critical that both new cells inherit an entire copy of the genetic material. This process is called mitosis, and it involves the duplicated chromosomes lining up in the middle of the cell before being pulled apart into the two newly forming cells. Many different proteins control mitosis because mistakes during cell division can lead to cells with too much or too little DNA, which can lead to cancers and other diseases.

Mitosis is mainly regulated by enzymes called kinases and phosphatases. Kinases add phosphate groups on to other proteins, which often changes their activity or localization within the cell. Phosphatases counteract the kinases by removing the phosphate groups. During mitosis, kinases and phosphatases accumulate at a specific region of the chromosomes called kinetochores. The kinetochores play two key roles: they are the regions from which the chromosomes are pulled apart and also serve as control centers for regulating the sequence of events in mitosis. To date, mitosis is best understood in yeast cells and less is known about the more complex process in human cells.

Previous research had shown that human cells need a group of proteins called the Ska complex to undergo mitosis, because without this complex the chromosomes remain in the middle of the cell and do not separate. Sivakumar et al. – who include two of the researchers involved in the previous research – have now explored the Ska complex's role in human cells in more detail. The experiments showed that the Ska complex binds to and recruits a phosphatase called PP1 to the chromosomes; this is not how PP1 is brought to the kinetochores in yeast. When enough PP1 is concentrated around the kinetochores this gives the human cell the signal to pull the chromosomes apart and finish mitosis.

Future work could ask how PP1 brings about the last stages of mitosis; for example, by finding all the proteins from which PP1 removes phosphate groups. Lastly, further studies could also explore the possibility that the Ska complex performs other tasks that are crucial for the division of human cells.

Centromere- and kinetochore-associated kinases play key roles in regulating chromosome attachment to the spindle and cell cycle progression in mitosis. As chromosomes align to the metaphase plate, PP1 becomes concentrated at kinetochores stabilizing kinetochore-microtubule interactions, and promoting anaphase onset by opposing spindle checkpoint kinase signaling (*Liu et al., 2010*; *Nijenhuis et al., 2014*). Recruitment of PP1 to the kinetochore then leads to substrate dephosphorylation, stabilization of kinetochore-microtubule attachments, and anaphase onset (*Liu et al., 2010*; *Nijenhuis et al., 2014*; *Meadows et al., 2011*; *Rosenberg et al., 2011*; *Zhang et al., 2014*).

The heterotrimeric spindle- and kinetochore-associated (Ska) protein complex consisting of Ska1–3 is required for timely anaphase onset (*Daum et al., 2009*; *Welburn et al., 2009*; *Gaitanos et al., 2009*). Some laboratories have reported that Ska depletion strongly compromises chromosome alignment (*Welburn et al., 2009*; *Gaitanos et al., 2009*; *Raaijmakers et al., 2009*). However, our detailed video analyses show that depletion of Ska components, individually or in combination, causes delays in chromosome alignment followed by a robust metaphase arrest (*Daum et al., 2009*; *Sivakumar et al., 2014*). Metaphase arrest is often then followed by cohesion fatigue and asynchronous chromatid separation without progression to anaphase (*Stevens et al., 2011*; *Daum et al., 2011*). Cells that have undergone cohesion fatigue remain arrested in mitosis with a terminal phenotype in which mixtures of separated chromatids and intact chromosomes are scattered across the spindle.

The C-terminal domain (CTD) of Ska1 binds to microtubules and has thus been termed the microtubule-binding domain or MTBD (*Abad et al., 2014*; *Schmidt et al., 2012*). HeLa cells expressing Ska1 lacking the CTD showed phenotypes similar to Ska depletion. Cells delayed at metaphase, and cold stable kinetochore fibers were reportedly decreased (*Abad et al., 2014*; *Schmidt et al., 2012*). Furthermore, in vitro the Ska complex was shown to track depolymerizing microtubule ends (*Schmidt et al., 2012*). These results have led to a model in which Ska directly promotes microtubule stability and kinetochore movement on microtubules in conjunction with other kinetochore components, such as the Ndc80 complex (*Abad et al., 2014*; *Schmidt et al., 2012*).

In this study, we report that Ska recruits PP1 to kinetochores. Compromising this recruitment increases phosphorylation of Knl1 and increases recruitment of the checkpoint kinase Bub1. We find that Ska1 CTD is required for binding to PP1 in vivo and in vitro. The metaphase arrest or delay seen after Ska depletion is strongly rescued by expressing a chimeric Ska1 protein with its CTD replaced either by a PP1-binding motif or by a direct fusion to PP1. Thus, a major function of the Ska complex is to recruit PP1 to the kinetochore.

## Results

### The SKA complex is required for kinetochore accumulation of PP1

Several mitotic kinases, including Mps1, Aurora B, Bub1 and Plk1 accrue to high levels at centromeres and kinetochores in early mitosis, during prophase and prometaphase (*Funabiki and Wynne, 2013*). An opposing phosphatase, PP1, also accumulates on kinetochores as microtubules attach, reaching maximal levels at metaphase (*Liu et al., 2010*; *Nijenhuis et al., 2014*; *Suijkerbuijk et al., 2012*; *Foley et al., 2011*). We tested if the recruitment of PP1 to kinetochores was affected in Ska3-depleted cells. We found that the kinetochore levels of PP1 were diminished in Ska3-depleted cells at prometaphase and metaphase (*Figure 1A*). In contrast, the pool of PP1 on the mitotic spindle appeared unaffected. Thus, the Ska complex is required to recruit PP1 to the kinetochore.

Since Ska depletion reduced PP1 recruitment to the kinetochore, we asked if overexpression of Ska would increase it. We had previously shown that expressing a fusion of the KMN component, Mis12, to Ska1 (Mis12-Ska1) increased total Ska recruitment to the kinetochore (*Sivakumar et al., 2014*). Cells expressing Mis12-Ska1 had higher levels of PP1 at kinetochores than control cells expressing either Mis12 or Ska1. The cells in this experiment were arrested at metaphase with the proteasome inhibitor, MG132 (*Figure 1B*). Expression of Mis12-Ska1 also increased PP1 levels at kinetochores in cells treated with nocodazole (*Figure 1C*), suggesting that targeting Ska complex to the kinetochore leads to accumulation of kinetochore PP1 in a microtubule-independent manner. Although expression of Mis12-Ska1 increased PP1 levels at the kinetochore, it was not sufficient to accelerate anaphase onset in normal cells or induce mitotic exit in nocodazole-treated cells (*Sivakumar et al., 2014*).

### The SKA complex physically interacts with PP1 through the SKA1 C terminal domain (CTD)

To better understand how the Ska complex recruits PP1 to the kinetochore, we first tested if the Ska complex physically interacted with PP1 in mitotic HeLa cell lysates. We found that Ska3 could indeed be co-immunoprecipitated (co-IPed) with PP1 (*Figure 2A*). As a positive control, Knl1, a kinetochore protein known to directly bind PP1 (*Liu et al., 2010*; *Meadows et al., 2011*; *Rosenberg et al., 2011*; *Zhang et al., 2014*), also co-IPed with PP1. To test whether the interaction between Ska and PP1 might be bridged by Knl1, we compared Ska3 IPs from control and Knl1-depleted cells (*Figure 2B*). The Ska complex interacted with similar amounts of PP1 in control and Knl1-depleted cells. Knl1 also co-IPed similar amounts of PP1 from control and Ska3-depleted cells, indicating that Ska depletion did not affect the Knl1-PP1 interaction (*Figure 2C*). By antibody staining and by using a HeLa-PP1γGFP stable cell line, we found that kinetochore PP1 levels were low and were slightly above the background signals of soluble, non-kinetochore PP1. Individual depletions of Knl1 or Ska3 reduced the kinetochore PP1 levels to the background levels. Thus, we were unable to determine the individual contributions of Ska and Knl1 toward kinetochore recruitment of PP1 by immunostaining. From our IP experiments, we conclude that the Ska complex and Knl1 are capable of independent binding to PP1.

To determine if the Ska complex and PP1 could interact in vitro and to analyze which Ska component was responsible for the interaction, we purified GST-Ska1/2 and GST-Ska3 proteins from bacteria and conducted binding assays with in vitro translated Myc-PP1. GST-Ska1/2 bound efficiently to Myc-PP1, with or without GST-Ska3 (*Figure 2D*). As positive controls, GST-Ska1/2 interacted with in vitro translated Ska3, and vice versa. Thus, Ska1/2 physically interacted with PP1 in vitro. We next tested whether GFP-Ska proteins could interact with PP1 in HeLa cells depleted of endogenous Ska components. We found that GFP-Ska1, but not GFP-Ska2 or GFP-Ska3, interacted with mCherry-PP1

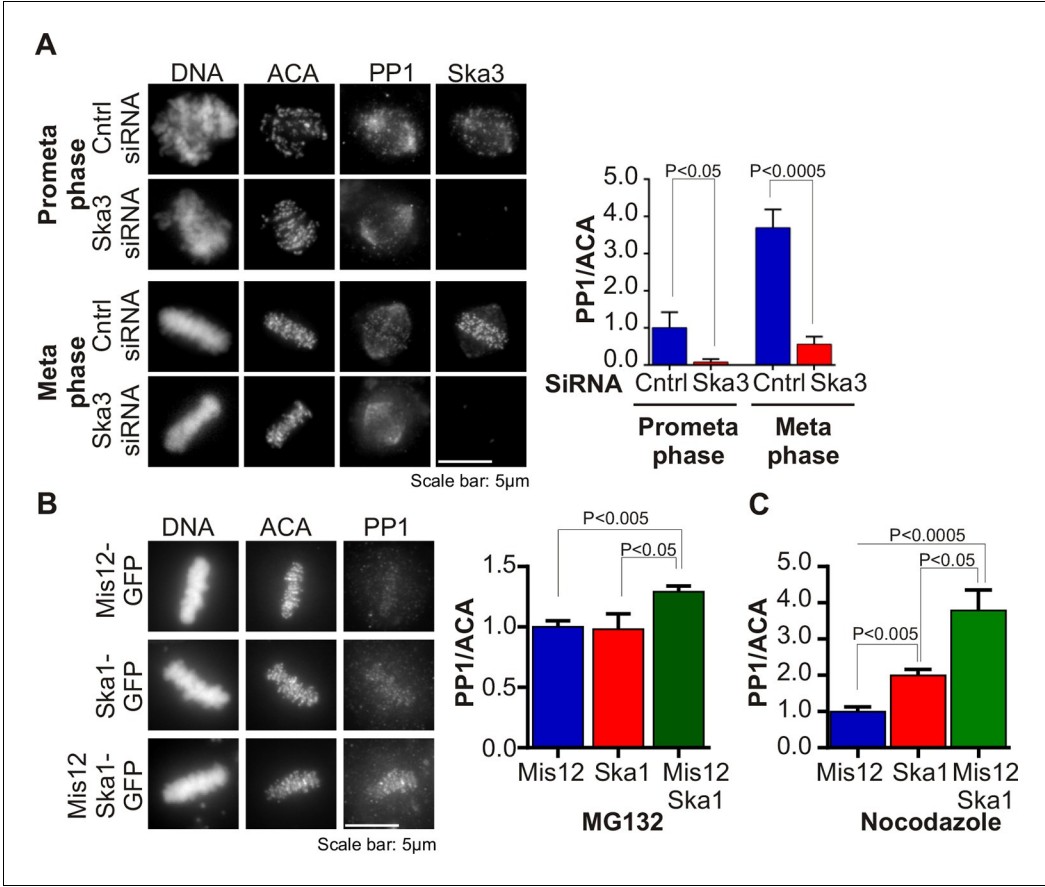

**Figure 1.** Ska complex is required for PP1 recruitment to the kinetochore. (A) HeLa cells were transfected with control or Ska3 siRNA at 50nM final concentration. Thirty hours after transfection immunofluorescence was done and PP1 at the kinetochore was quantified. Ska3 antibody staining shows efficiency of depletion. PP1 at kinetochores increases from prometaphase to metaphase. Ska3-depleted cells are inefficient in PP1 recruitment to kinetochores in both prometaphase and metaphase. (B) HeLa cells were transfected with Mis12-GFP, Ska1-GFP or Mis12Ska1-GFP to increase Ska complex accumulation at kinetochores. Thirty-six hours after transfection, MG132 was added for 1 hr to accumulate cells at metaphase. Immunofluorescence of PP1 at kinetochores was quantified. PP1 accumulates at kinetochores in Mis12Ska1GFP-expressing cells to a greater extent than in Mis12GFP- or Ska1GFP-expressing cells. (C) In cells treated with 3.3 µM nocodazole, PP1 accumulated to higher levels at kinetochores of cells expressing Mis12Ska1-GFP compared to cells expressing Mis12GFP or Ska1GFP.

in lysates prepared from HeLa cells depleted of the other two endogenous Ska components (*Figure 2E*). Thus, PP1 binding by the Ska complex is mainly mediated by Ska1.

The N-terminal coiled coil region of Ska1 is required to form a complex with Ska2 and Ska3, whereas its CTD has been well characterized as a microtubule-binding domain (*Abad et al., 2014*; *Schmidt et al., 2012*). To determine which region of Ska1 was responsible for binding PP1, we performed in vitro binding assays with bacterially purified GST-tagged fragments of Ska1 and in vitro translated Myc-PP1. The Ska1 CTD was found to interact with PP1 (*Figure 3A*). To further test if PP1 and Ska complex bind each other directly, gel-filtration experiments were carried out with purified, bacterially expressed Ska and PP1 proteins. The Ska complex containing full-length Ska1 bound PP1, while the Ska complex containing Ska1 lacking the CTD showed reduced complex formation (*Figure 3B*). Using microscale thermophoresis, we found that purified recombinant Ska1 CTD and PP1 proteins interacted with a $K_d$ of 1.5 µM (*Figure 3C*, *Figure 3—figure supplement 1*). Thus, Ska1 CTD and PP1 directly interact with moderate affinity. Other parts of Ska or associated proteins may enhance the Ska1 CTD-PP1 interaction in vivo.

In HeLa cells depleted of endogenous Ska complex components and expressing RNAi-resistant Ska1 fragments, GFP-Ska1 CTD bound to Myc-PP1 (*Figure 4A*). We then constructed doxycycline

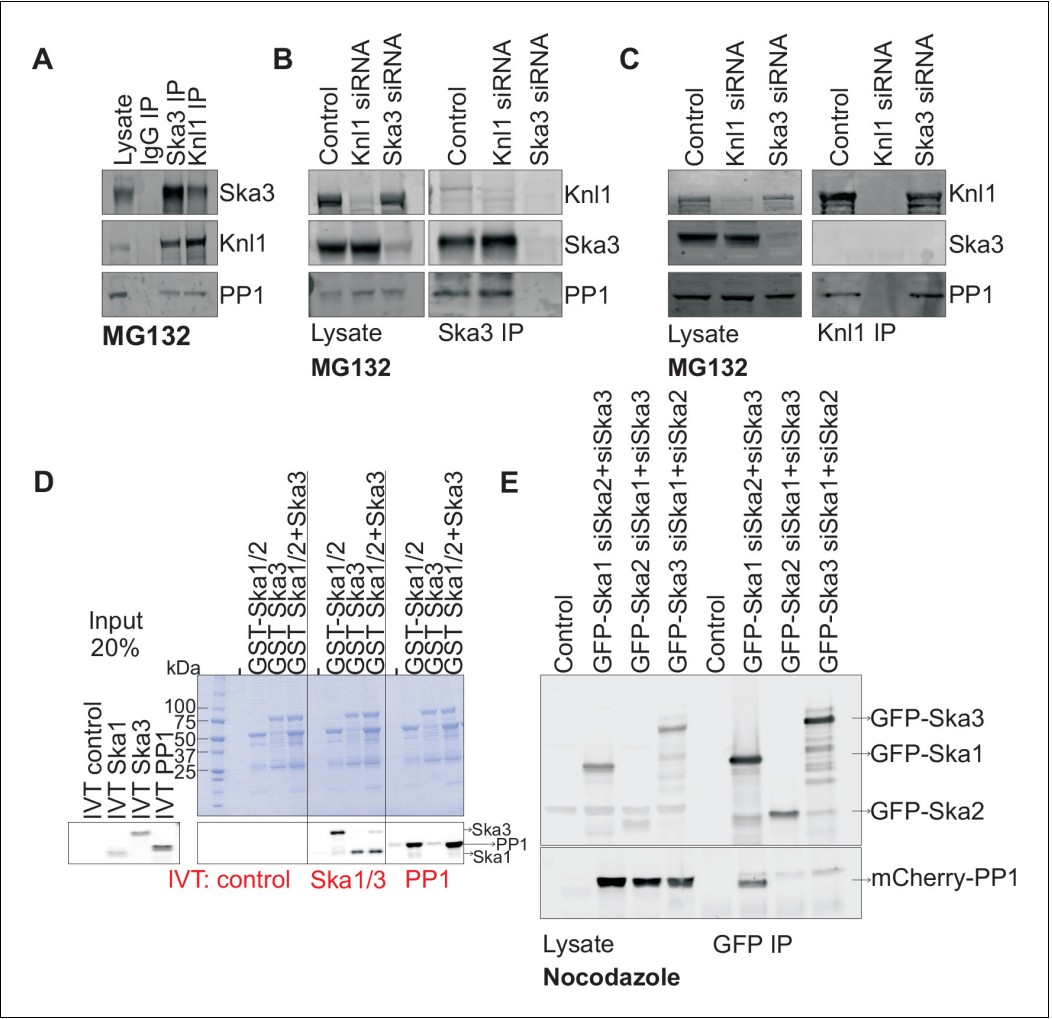

**Figure 2.** Ska1 interacts with PP1. (**A**) HeLa cells synchronized in S phase with thymidine were then released and arrested in mitosis using 0.33 μM nocodazole. The cells were subsequently released from nocodazole into MG132 (25 μM) for 1 hr to allow metaphase chromosome alignment. Cell pellets were lysed and IgG, Ska3, and Knl1 IPs were performed to detect interaction with PP1. Both Ska3 and Knl1 IPs, but not control IgG IP, pulled down PP1 from cell lysates. (**B**) HeLa cells were transfected with mock siRNA, Knl1 siRNA, or Ska3 siRNA. HeLa cells were then arrested at metaphase as decribed above in A. Ska3 co-precipitated with similar amounts of PP1 in extracts prepared from control and Knl1-depleted cells. (**C**) HeLa cells transfected with mock, Knl1, or Ska3 siRNA were synchronized at metaphase as described above in A. Knl1 co-precipitated with similar amounts of PP1 in extracts prepared from control and Ska3-depleted cells. (**D**) GST-Ska proteins purified from bacteria were incubated with in vitro translated S$^{35}$-labeled Myc-PP1. GST-Ska1/2 and GST Ska1/2+GST Ska3 interacted with S$^{35}$-labeled Myc-PP1. E. HeLa cells were transfected with mCherry-PP1 and control, GFP Ska1, GFP Ska2 or GFP Ska3 plasmids. Endogenous Ska components were depleted using siRNA; cells were synchronized in S phase using thymidine then released and arrested in mitosis using 3.3 μM nocodazole. GFP-Ska1 precipitated from cells depleted of Ska2 and Ska3 showed the strongest interaction with mCherry-PP1.

(Dox)-inducible HeLa cell lines stably expressing RNAi-resistant, GFP fusions of full-length Ska1 or Ska1ΔCTD. As reported previously (*Abad et al., 2014*; *Schmidt et al., 2012*), exogenously expressed full-length Ska1 localized to centrosomes, spindle microtubules, and kinetochores during mitosis while Ska1ΔCTD localized predominantly to kinetochores (*Figure 4—figure supplement 1*). We arrested these cells at metaphase with MG132 and immunoprecipitated the full-length Ska1 or Ska1ΔCTD with anti-GFP antibody. Full-length Ska1 interacted more efficiently with endogenous PP1 than did Ska1ΔCTD, with or without depletion of endogenous Ska1 (*Figure 4B*). Additionally, when we overexpressed Myc-PP1 in the stable cell lines, we detected an interaction between Ska1

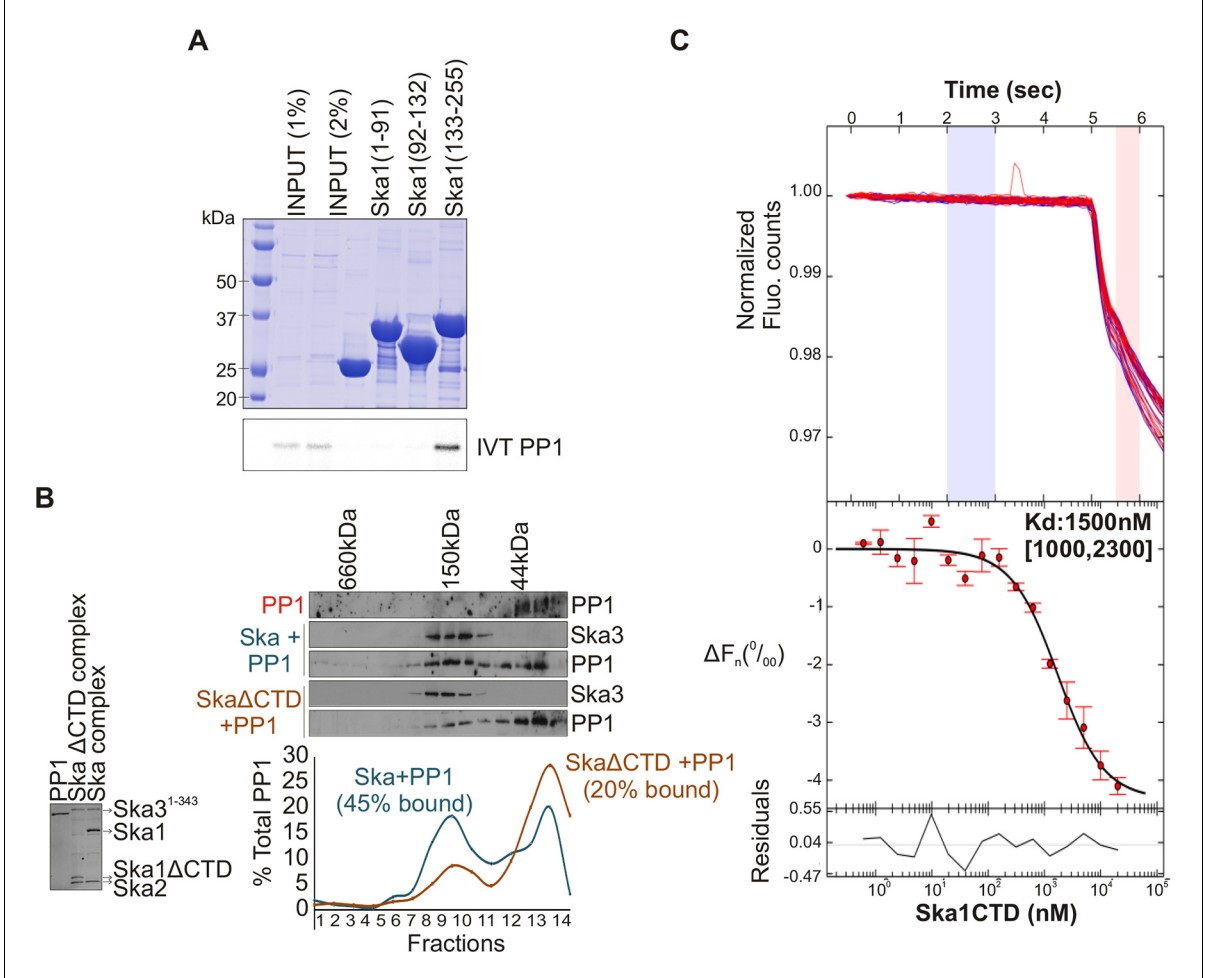

**Figure 3.** Ska1 C terminal domain (CTD) physically interacts with PP1. (**A**) GST fusions to Ska1 fragments, GST-Ska1(1–91), GST-Ska1(92–132) and GST-Ska1(133–255) were purified from bacteria. In vitro translated S[35] labeled Myc-PP1 was added to GST-Ska1 fragments and in vitro binding assays were performed to detect Myc-PP1 interaction with GST-Ska1 fragments. GST-Ska1(133–255) bound strongly with Myc-PP1 while other fragments did not. (**B**) (Left) Purified 6xHis-PP1α$^{7-330}$ (PP1), untagged Ska1ΔCTD/Ska2/Ska3$^{1-343}$ and Ska1$^{FL}$/Ska2/Ska3$^{1-343}$, visualized on coommassie stained gel. (Right) Immunoblots showing the elution profile of PP1, PP1 incubated with Ska1$^{FL}$/Ska2/Ska3$^{1-343}$ (blue) or PP1 incubated with Ska1ΔCTD/Ska2/Ska3$^{1-343}$ (orange) run on a Superose 12 size-exclusion column. Ska complexes run in the same fraction in the absence of PP1 (not shown). Densitometry quantifications of PP1 signal in fractions eluted from size-exclusion column shows a reduction in binding to Ska complex lacking the Ska1 CTD. (**C**) Microscale thermophoresis (MST) was done to analyze the direct binding interaction between PP1 and the Ska1 CTD proteins purified from bacteria. The top panel shows thermophoretic time traces of 16 samples in three independent experiments. The middle panel shows the T-Jump data (circles) and the fit to the data (line). The residuals between the data and the fit line are indicated in the bottom panel. The $K_d$ of PP1 binding to the Ska1 CTD was calculated to be 1.5 μM. $F_n$ represents ratio (expressed in per-mille units) of the fluorescence readings in the time traces as measured just after (pink region, top panel) and before (blue region, top panel) activation of the MST laser; $\Delta F_n$ is calculated by subtracting the refined $F_n$ of the free PP1 from all $F_n$ values and thus represents the change in T-Jump response as a function of ligand concentration. This part was rendered using the program GUSSI (*Brautigam, 2015*).

The following figure supplement is available for figure 3:

**Figure supplement 1.** Purification of PP1 and Ska1CTD from bacteria His-PP1γ and GST-Ska1CTD were individually purified from bacteria.

and Myc-PP1 even in nocodazole-treated cells where association of Ska with endogenous PP1 is difficult to detect, perhaps due to the reduced recruitment of Ska to kinetochores in mitotic cells lacking microtubules (*Figure 4C*). Again, Ska1ΔCTD showed reduced Myc-PP1 binding both in nocodazole- and in MG132-treated cells (*Figure 4C-figure supplement 2*).

The Ska1 CTD does not contain clear matches for the known PP1-binding motifs present in many PP1-interacting proteins. We thus made a series of point mutants targeting surface-exposed,

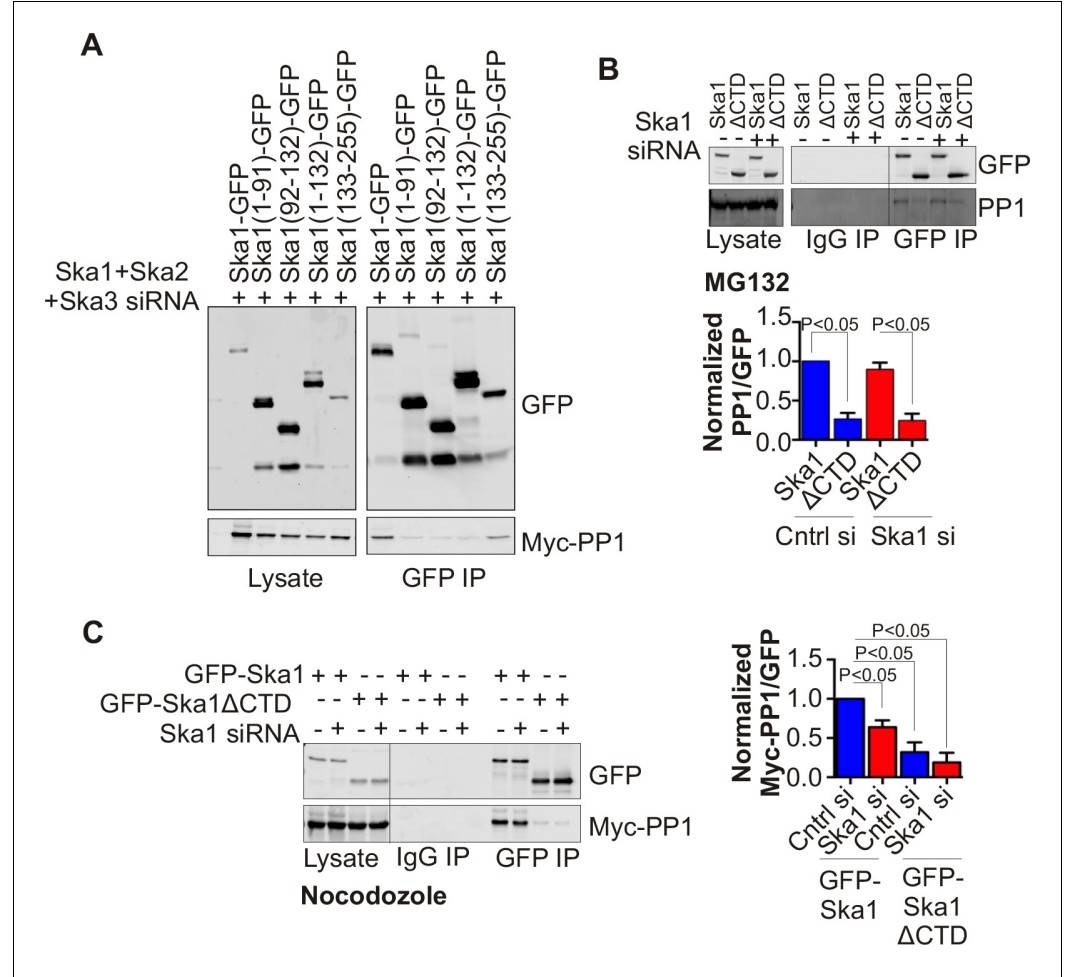

**Figure 4.** Ska1CTD immunoprecipitates with PP1 in HeLa cells. (**A**) HeLa cells were transfected with GFP-Ska1(1–91), GFP-Ska1(92–132), GFP-Ska1(1–132), GFP-Ska1(133–255), and Myc-PP1 plasmids. Endogenous Ska1, Ska2, and Ska3 were depleted with siRNA. GFP IP's followed by Western blotting was done to analyze the GFP-Ska1 fragment that associated most strongly with Myc-PP1 in vivo. GFP-Ska1(133–255) interacted with Myc-PP1 in extracts from cells depleted of all endogenous Ska components. (**B**) HeLa cell lines expressing GFP-Ska1 or GFP-Ska1ΔCTD under Doxocycline (Dox) control and resistant to siRNA were generated. These were transfected with control or Ska1 siRNA and transgene expression was induced. Cells were synchronized by thymidine in S phase then released and arrested in mitosis using 0.33 µM nocodazole. Mitotic cells were released into MG132 (25 µM) to allow metaphase chromosome alignment and collected after 1 hr. GFP IP's were prepared from the cell extracts and the amount of PP1 associated determined by quantitative immunoblotting. IP of GFP-Ska1ΔCTD showed greatly reduced binding to PP1 both with or without depletion of endogenous Ska1. (**C**) Dox-inducible HeLa GFP-Ska1 and GFP-Ska1ΔCTD cells were transfected with Myc-PP1 plasmid followed 6 hr later by transfection of control or Ska1 siRNA. Dox was added to induce transgene expression. Cells were synchronized in S phase with thymidine then released and collected in mitosis using 3.3 µM nocodazole. GFP IP's were done followed by quantitative immunoblotting to detect Myc-PP1 association. GFP-Ska1 coprecipitated significant amonts of Myc-PP1, while GFP-Ska1ΔCTD (with or without depletion of endogenous Ska1) showed greatly reduced co-precipitatation with Myc-PP1. Depletion of endogenous Ska1 did reduce somewhat the ability of full length GFP-Ska1 to co-precipitate Myc-PP1. However, GFP-Ska1ΔCTD showed pronounced reduction in co-precipitation of Myc-PP1.

The following figure supplements are available for figure 4:

**Figure supplement 1.** Characterization of HeLa Tet-On cells expressing GFP-Ska1 or GFP-Ska1ΔCTD HeLa cells stably transfected with constructs for inducible expression of GFP-Ska1 or GFP Ska1ΔCTD cells were treated with Dox to induce transgene expression.

*Figure 4 continued on next page*

*Figure 4 continued*

**Figure supplement 2.** GFP-Ska1ΔCTD shows reduced binding to Myc-PP1 in extracts from cells arrested at metaphase with MG132..

**Figure supplement 3.** Characterization of Ska1CTD to identify residues involved in interaction with PP1.

conserved residues in Ska1 CTD, and systematically tested their interaction with Myc-PP1 in nocodazole-treated HeLa cells upon endogenous Ska1 depletion. We were unable to identify point mutants that disrupted the interaction between Ska1 and PP1 (*Figure 4—figure supplement 3*).

## The SKA1 CTD is required for PP1 kinetochore targeting and KNL1 dephosphorylation

Since Ska1ΔCTD was deficient in binding PP1, we expected the cell line expressing this mutant to show reduced kinetochore localization of PP1. To eliminate potential PP1 binding to microtubules or microtubule-associated proteins, we treated the cells with high concentration (3.3 µM) nocodazole. PP1 localization to the kinetochore was decreased by 50% in cells expressing Ska1ΔCTD, compared to cells expressing full-length Ska1 (*Figure 5A*), with or without depletion of Ska1. Thus, Ska1ΔCTD could diminish PP1 kinetochore targeting in a dominant negative fashion, confirming a role for the Ska1 CTD in PP1 targeting to kinetochores.

The spindle checkpoint kinase Mps1 phosphorylates Knl1 on multiple MELT motifs, and the phosphorylated MELT recruits other checkpoint components, such as Bub1, to kinetochores (*Yamagishi et al., 2012*; *Shepperd et al., 2012*; *London et al., 2012*). We found that phosphorylation of one such MELT motif, pMELT (residue pT875), was indeed elevated in cells expressing Ska1ΔCTD, as compared to cells expressing full-length Ska1 (*Figure 5B*) (*Ji et al., 2015*). Consistent with this finding, cells expressing Ska1ΔCTD showed increased levels of Bub1 protein at kinetochores, as compared to cells expressing full-length Ska1 (*Figure 5C*). Differences in PP1, pMELT and Bub1 labeling were not simply due to differences in expression levels in the stable cell lines, since quantification showed that GFP signals in GFP-Ska1 and GFP-Ska1ΔCTD cells were similar (*Figure 5—figure supplement 1*). These results indicate that PP1 recruited by the Ska1 CTD opposes Knl1 phosphorylation by Mps1 directly or indirectly.

## SKA-mediated kinetochore recruitment of PP1 promotes anaphase onset

PP1 is recruited to the kinetochore to dephosphorylate kinetochore proteins to oppose spindle checkpoint signaling and promote anaphase onset. PP1 depletion was found to cause delays at metaphase (*Liu et al., 2010*; *Nijenhuis et al., 2014*; *Zhang et al., 2014*). Our experiments show that cells expressing Ska1ΔCTD are deficient in PP1 recruitment to the kinetochore. We sought to determine whether the kinetochore recruitment of PP1 by Ska affected mitotic progression. Depletion of Ska proteins by RNAi causes long metaphase delays or arrest (*Sivakumar et al., 2014*). As expected, expression of RNAi-resistant Ska1 rescued the arrest caused by Ska1 depletion (*Figure 6A*). Consistent with previous reports (*Abad et al., 2014*; *Schmidt et al., 2012*), expression of Ska1ΔCTD did not rescue the metaphase arrest/delay caused by Ska1 depletion (*Figure 6A,B*).

We then tested if Ska1ΔCTD fused to a PP1 binding motif rescued timely anaphase onset. We expressed a chimeric protein consisting of Ska1ΔCTD fused to residues 34–81 of Knl1. This region of Knl1 contains the PP1-binding RVSF motif, but importantly it lacks the microtubule-binding domain found at the extreme N terminus of Knl1. Expression of Ska1ΔCTD-Knl1[(34–81)] rescued anaphase onset in cells depleted of endogenous Ska1 (*Figure 6A–C*). Mutation of the RVSF motif to AAAA was previously shown to block binding of PP1 to Knl1 (*Liu et al., 2010*). Expression of Ska1ΔCTD fused to the Knl1 fragment with the RVSF to AAAA mutation failed to rescue metaphase arrest/delay in the majority of cells depleted of endogeneous Ska1. This result suggests that the decreased PP1 recruitment contributes to the Ska depletion phenotype (*Figure 6A–C*).

To further test if direct PP1 recruitment by Ska1 could promote anaphase onset, we created a chimeric Ska1 protein with its CTD replaced with PP1. Gratifyingly, expression of the Ska1-PP1 chimeric protein produced higher PP1 accumulation (1.5-fold increase) at kinetochores (*Figure 7—figure*

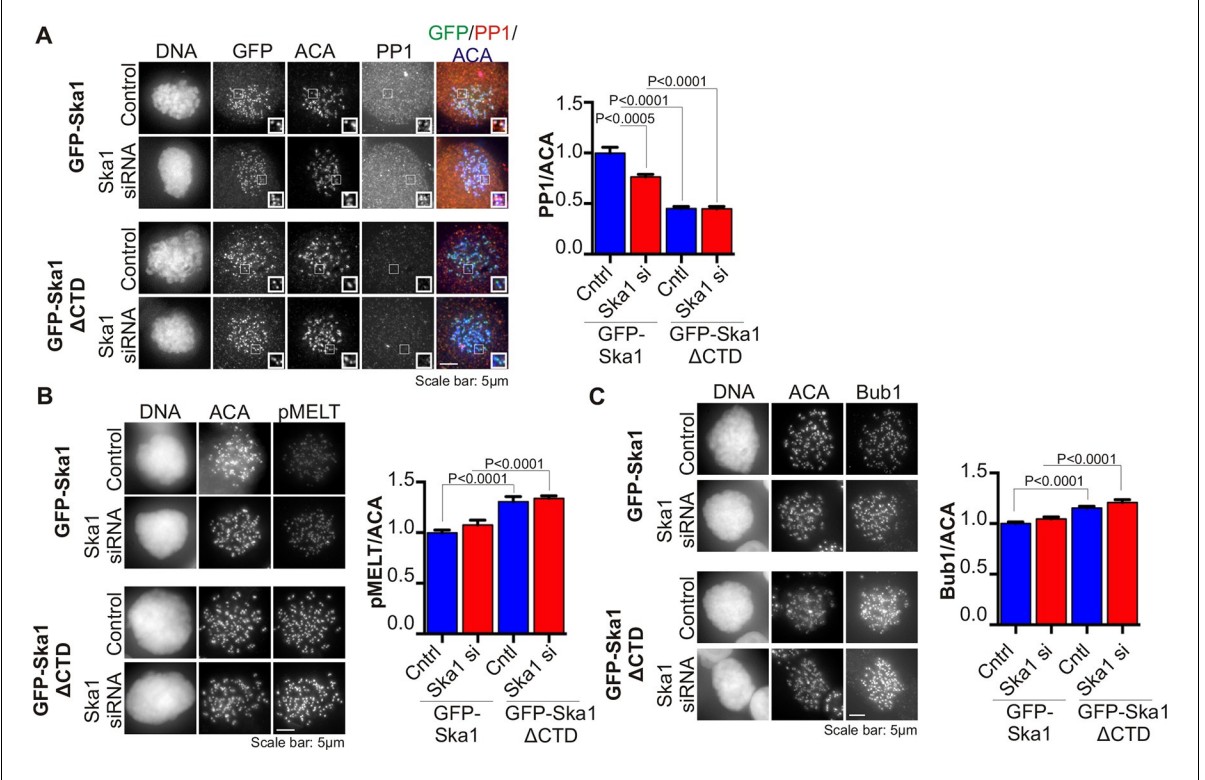

**Figure 5.** Cells expressing Ska1ΔCTD recruit less PP1 and accumulate more Knl1 phosphoepitope and Bub1 protein at kinetochores. HeLa GFP-Ska1 and HeLa GFP-Ska1ΔCTD cells were grown on chambered cover slides. Dox was added to induce transgene expression. Cells were transfected with Ska1 siRNA, and thymidine was added for 18–24 hr to synchronize cells. Cells were released from thymidine and arrested in mitosis using 3.3 µM nocodazole. Immunofluorescence was done to detect PP1, pMELT, Bub1 at the kinetochore. (**A**) Image panel showing PP1 localization at the kinetochore in GFP-Ska1 and GFP-Ska1ΔCTD cells. The graph depicts the decrease in PP1 localization in GFP-Ska1ΔCTD cells compared to GFP-Ska1 cells both with and without depletion of endogenous Ska1. (**B**) Images show localization of antibody (pMELT) to a phosphoepitope on Knl1 (pT875) at kinetochores in GFP-Ska1 and GFP-Ska1ΔCTD cells. The graph shows that cells expressing GFP-Ska1ΔCTD have increased pMELT signals at kinetochores compared to GFP-Ska1-expressing cells. (**C**) Images show the levels of Bub1 protein at kinetochores in GFP-Ska1 and GFP-Ska1ΔCTD cells. Qantification shows that GFP-Ska1ΔCTD accumulate more Bub1 at kinetochores compared to GFP-Ska1 cells.

The following figure supplement is available for figure 5:

**Figure supplement 1.** Expression levels of cell lines stably expressing inducible GFP-Ska1 and GFP-Ska1ΔCTD are similar After induction with Dox, GFP expression levels in individual cells were quantified.

supplement 1), and resulted in complete rescue of the mitotic arrest and alignment delay phenotypes and substantially reduced the metaphase delay phenotype induced by Ska1 depletion (**Figure 7A**). As a control, expression of PP1 alone, not fused to Ska1, failed to rescue. Thus, expression of a Ska1-PP1 chimeric protein, lacking the CTD and its associated microtubule-binding activity, significantly alleviates the mitotic phenotypes caused by Ska1 depletion. As shown previously (**Schmidt et al., 2012**) and confirmed here, expression of Ska1ΔCTD caused metaphase delays even in cells without Ska1 depletion, indicating that it dominant negatively inhibits Ska function (**Figure 7B**). This dominant-negative phenotype was not seen with the Ska1-PP1 chimeric protein in which the CTD was replaced by PP1 (**Figure 7A,B—figure supplement 2**). These results are consistent with a role for Ska1-recruited PP1 in mitotic progression.

To determine if the phosphatase activity of PP1 was essential for the rescue of Ska1-depleted cells expressing the Ska1-PP1 chimera, we generated an expression vector containing a fusion protein with a point mutation (H248K) in PP1 that abolished its catalytic activity (**Hirschi et al., 2010**). Expression of PP1 H248K (phosphatase dead PP1/pdPP1) without fusion to Ska1 was toxic to cells and induced cell death in interphase (data not shown). Expression of Ska1ΔCTD-PP1 H248K

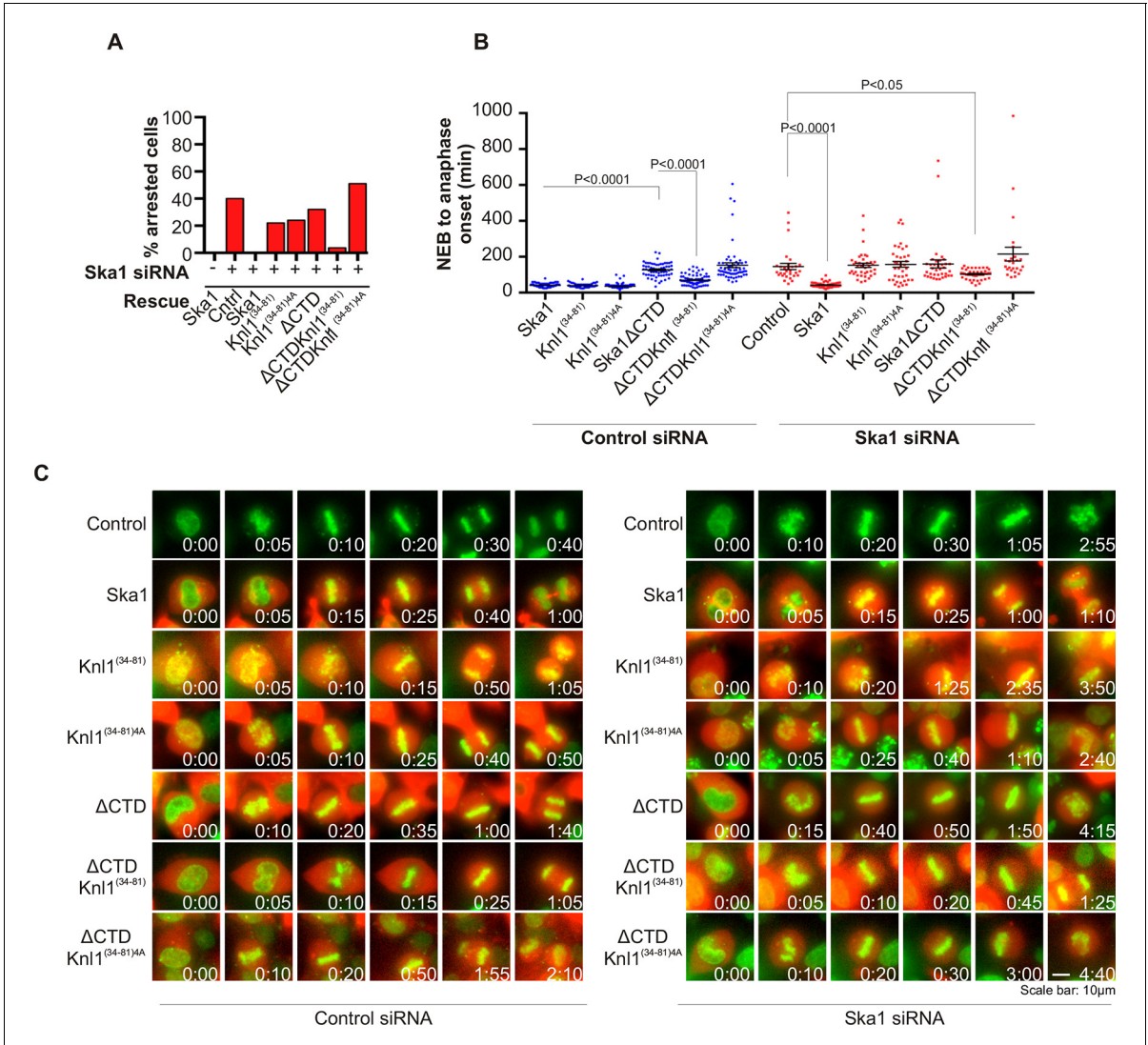

**Figure 6.** Expression of Ska1ΔCTD fused to the PP1-binding domain (amino acids 34–81) of Knl1 rescues phenotypes caused by Ska1 depletion. (**A**) HeLa cells were transfected with plasmids to express mCherry-Ska1, mCherry-Knl1[(34–81)], mCherry-Knl1[(34–81)4A] (where the PP1 binding motif RVSF is mutated to AAAA), mCherry-Ska1ΔCTD, mCherry-Ska1ΔCTD fused to Knl1[(34–81)] (mCherry-Ska1ΔCTDKnl1[(34–81)]) and mCherry-Ska1ΔCTDKnl1[(34–81)4A]. Endogenous Ska1 was depleted using Ska1 siRNA. Cells were imaged by time-lapse microscopy, and % of mitotic cells arrested in metaphase was plotted. As expected, expression of siRNA-resistant mCherry-Ska1 rescued metaphase arrest caused by Ska1 depletion while expression of mCherry-Ska1ΔCTD did not. Expression of Ska1ΔCTDKnl1[(34–81)] fusion in Ska1-depleted cells rescued metaphase arrest while expression of Ska1ΔCTDKnl1[(34–81)4A] did not. (**B**) The interval from NEB to anaphase onset is plotted for cells that progressed to anaphase in cells expressing Ska1 constructs with or without depletion of endogenous Ska1. In cells not depleted of endogenous Ska1, dominant negative effects in delaying mitotic progression were caused by expression of Ska1ΔCTD, as previously reported, and by expression of the fusion Ska1ΔCTDKnl1[(34–81)4A], which lacks the PP1-binding motif. Expression of other constructs including Ska1ΔCTDKnl1[(34–81)] with intact PP1 binding caused no delay. In cells depleted of endogenous Ska1, 40% of cells arrested at metaphase and did not progress to anaphase (*Figure 6A*). The rest showed delayed mitotic progression with an average of 146 min from NEB to anaphase onset compared to control cells (40 min). As expected the delay was rescued by expression of mCherry-Ska1 but was not rescued by expression of mCherry-Knl1[(34–81)], mCherry-Knl1[(34–81)4A] or mCherry-Ska1ΔCTD. Expression of Ska1ΔCTDKnl1[(34–81)] but not Ska1ΔCTDKnl1[(34–81)4A] showed significant rescue of the delay caused by Ska1 depletion. (**C**) Representative images of cells transfected with the indicated constructs treated with control siRNA or Ska1 siRNA. The green and red colors indicate DNA and mCherry expression, respectively. Only cells expressing indicated mCherry constructs were analyzed for this experiment. The time is indicated in hour:minutes.

(Ska1ΔCTDpdPP1) allowed cells to enter mitosis, but failed to rescue the mitotic delay of Ska1-depleted cells (*Figure 7C,D,E—figure supplement 2*). The chromosome alignment delays observed upon Ska1 depletion were also rescued by expression of Ska1-PP1 fusion protein but not by

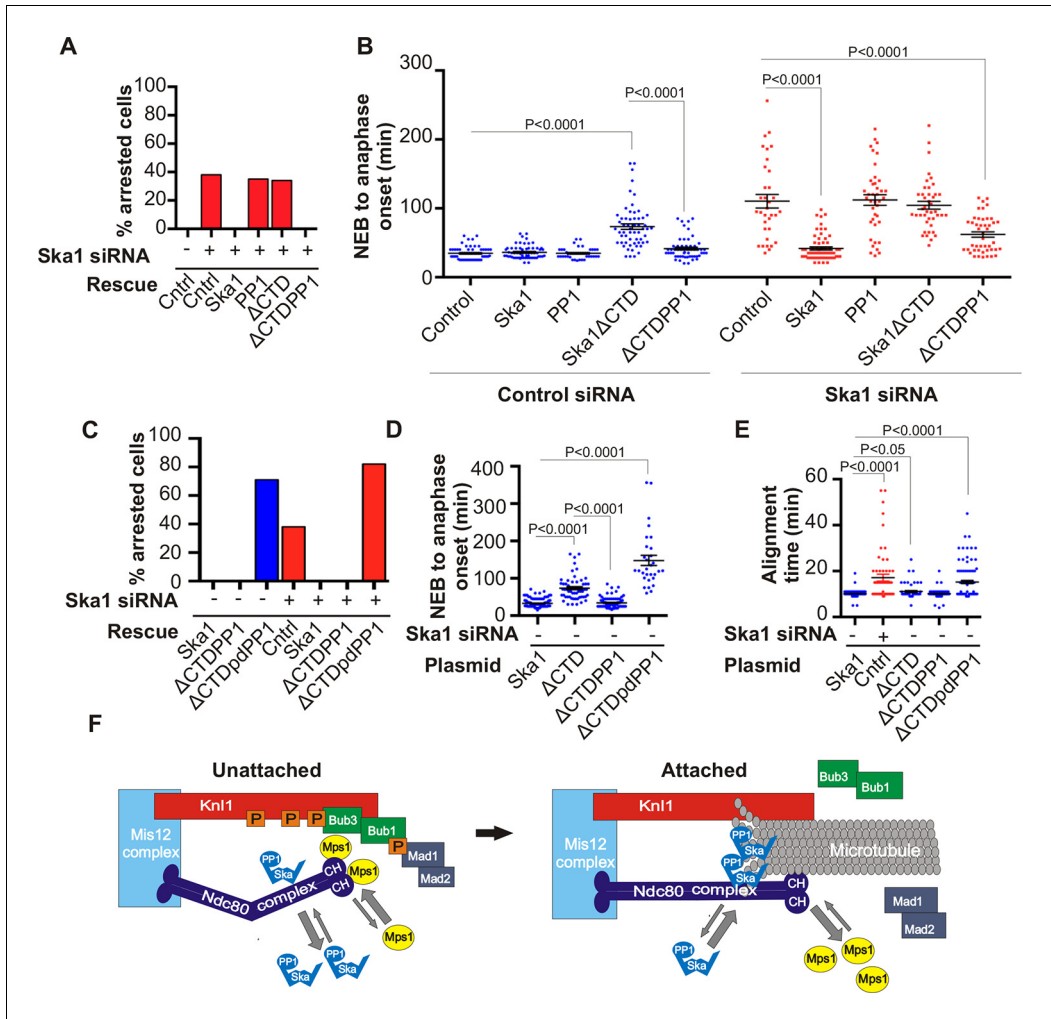

**Figure 7.** Expression of Ska1ΔCTD fused directly to PP1 but not phosphatase-dead PP1 (pdPP1) rescues phenotypes caused by Ska1 depletion. (**A**) HeLa cells were transfected with plasmids to express mCherry-Ska1, mCherry-PP1, mCherry-Ska1ΔCTD, and mCherry-Ska1ΔCTD fused to PP1 (mCherry-Ska1ΔCTDPP1). Endogenous Ska1 was depleted using Ska1 siRNA. Hoechst 33342 was added at 25 ng/ml to visualize DNA. Cells were then imaged by time-lapse microscopy, and % of mitotic cells arrested in metaphase was plotted. As expected, expression of siRNA-resistant mCherry-Ska1 rescued metaphase arrest caused by Ska1 depletion while expression of mCherry-PP1 or mCherry-Ska1ΔCTD did not. Expression of Ska1ΔCTDPP1 fusion in Ska1 depleted cells completely rescued metaphase arrest. (**B**) The interval from NEB to anaphase onset is plotted for cells that progressed to anaphase while expressing Ska1 constructs without or with depletion of endogenous Ska1. As expected, Ska1ΔCTD-expression showed a dominant negative effect delaying mitotic progression in control cells not depleted of endogenous Ska1. Expression of the fusion, Ska1ΔCTDPP1, caused no delay. When endogenous Ska1 was depleted, 38% of cells arrested at metaphase and did not progress to anaphase (**Figure 7A**). The rest showed delayed progression from NEB to anaphase with an average of 110 min compared to control cells (35 min). As expected, the delay was rescued by expression of mCherry-Ska1 but was not rescued by expression of mCherry-PP1 or mCherry-Ska1ΔCTD. Expression of Ska1ΔCTDPP1 showed significant rescue of the delay cause by Ska1 depletion with an average time from NEB to anaphase of 62 min. (**C**) HeLa cells were transfected with the indicated plasmids and then treated with mock or Ska1 siRNA. Expression of Ska1ΔCTD fused to a phosphatase dead PP1 (Ska1ΔCTDpdPP1) failed to rescue Ska1 depletion. Indeed, expression of phosphatase dead fusion, on its own, induced a potent metaphase arrest phenotype in cells not depleted of endogenous Ska1. Moreover, it exacerbated the metaphase arrest in cells depleted of Ska1. (**D**) Ska1ΔCTDpdPP1 causes a longer delay to anaphase onset than Ska1ΔCTD even without depletion of endogenous Ska1. (**E**) Chromosome alignment is delayed in Ska1 depleted cells and this is recapitulated by expression of Ska1ΔCTDpdPP1 without endogenous Ska1 depletion. Delays in chromosome alignment are not observed upon expression of Ska1ΔCTDPP1 fusion. (**F**) Hypothetical model for dynamic balance of Mps1 kinase and PP1 phosphatase activities during mitotic

*Figure 7 continued on next page*

*Figure 7 continued*

progression. Photobleaching studies have shown that Mps1, Ska, and PP1 all exhibit high turnover at kinetochores with a residence times of a few seconds (*Raaijmakers et al., 2009*; *Howell et al., 2004*; *Trinkle-Mulcahy et al., 2003*). Before microtubule attachment, Mps1 concentration at kinetochores remains high due to interaction with the CH domains of the Ndc80 complex. Correspondingly, Ska-PP1 concentrations are low because of the paucity of microtubules. The high Mps1 and low PP1 concentrations maintain high phosphorylation Mps1 substrates, Knl1 and Bub1. Microtubules compete with Mps1 for binding to the CH domains of the Ndc80 complex, resulting in depletion of Mps1. The binding of Ska to microtubule protofilaments increases PP1 concentration. High Ska-PP1 and low Mps1 result in dephosphorylation of substrates, promoting release of Bub1–Bub3 and Mad1–Mad2 complexes. Diminished checkpoint signaling due to release of Bub1–Bub3 and Mad1–Mad2 from kinetochores promotes anaphase onset and mitotic exit.

The following figure supplements are available for figure 7:

**Figure supplement 1.** Ska1ΔCTDPP1 fusion increases PP1 concentration at kinetochores.

**Figure supplement 2.** Ska1ΔCTDPP1 fusion but not Ska1ΔCTDpdPP1 (pdPP1-phosphatase dead PP1) rescues mitotic phenotypes observed upon Ska1 depletion.

---

expression of Ska1ΔCTDpdPP1 (*Figure 7E*). Indeed, mere expression of Ska1ΔCTDpdPP1 without Ska1 depletion resulted in phenotypes remarkably similar to those caused by Ska1 depletion, indicating that this mutant acted in a dominant negative manner. These data indicate that a major function of the Ska complex is to recruit active PP1 to kinetochores.

## Discussion

Before the alignment of chromosomes at the metaphase plate, the spindle checkpoint inhibits anaphase onset through signals generated by protein interactions at unattached kinetochores. This signaling requires the KMN complex, composed of the Knl1 protein, the Mis12 protein complex, and the Ndc80 protein complex. The central checkpoint kinase, Mps1, phosphorylates several methionine-glutamate-leucine-threonine (MELT) motifs on Knl1. The Bub1–Bub3 complex binds to phosphorylated MELT motifs and further recruits its binding partner, the BubR1–Bub3 complex (*Yamagishi et al., 2012*; *Shepperd et al., 2012*; *London et al., 2012*; *Primorac et al., 2013*). Mps1 also phosphorylates Bub1 to promote recruitment of the Mad1–Mad2 protein complex (*Yamagishi et al., 2012*; *London et al., 2012*). In the absence of microtubules, Mps1 associates with kinetochores via the Calponin homology (CH) domains of the Ndc80 complex (*Hiruma et al., 2015*; *Ji et al., 2015*; *Kemmler et al., 2009*; *Nijenhuis et al., 2013*; *Zhu et al., 2013*). Photobleaching studies show that Mps1 binding to kinetochores is highly dynamic, with a turnover time of ~13 s (*Howell et al., 2004*). The binding of microtubules to the Ndc80 complex displaces Mps1, and in budding yeast, this results in a rearrangement of kinetochore substructure such that Mps1 loses access to Knl1 (*Aravamudhan et al., 2015*). In metazoans, microtubule binding to Ndc80 displaces Mps1 from kinetochores (*Hiruma et al., 2015*; *Ji et al., 2015*). Thus, displacement of Mps1 from the Ndc80 complex by microtubule binding appears to be a central element of turning off kinase signaling. In metazoans, dynein 'stripping' of checkpoint proteins, including Mad1, Mad2 and BubR1, plays an additional role in down regulating the checkpoint signal.

The most prominent phenotype of Ska complex depletion is arrest or long delay at metaphase, and these phenotypes require intact spindle checkpoint signaling. Abrogating checkpoint signaling with a chemical inhibitor of Mps1 induces mitotic exit in cells arrested in mitosis with high concentrations of microtubule poisons, but this exit is slower in cells depleted of Ska (*Sivakumar et al., 2014*). This result is consistent with the idea that Ska functions in opposition to and downstream of checkpoint signaling. Here, we show that the Ska complex is required for full kinetochore recruitment of PP1, a likely candidate for reversing checkpoint kinase phosphorylations. The fact that Ska depletion cannot completely block mitotic exit when Mps1 inhibitors are added likely reflects the fact that other pools of PP1 or other phosphatases, particularly PP2A, may also play a role in mitotic exit (*Nijenhuis et al., 2014*; *Foley and Kapoor, 2013*; *Grallert et al., 2015*). However, the strong

metaphase arrest phenotype seen after Ska depletion attests to the importance of Ska-associated PP1 in regulating the metaphase-anaphase transition in normal mitosis.

We found that the Ska1 protein, and specifically the Ska1 CTD, previously shown to bind microtubules, is essential for binding PP1 and recruiting it to the kinetochore. Since the C-terminal domain is involved in binding to both PP1 and microtubules, we propose to name this domain simply as the CTD, instead of the MTBD. Consistent with a role in opposing Mps1 checkpoint signaling, we find that in cells where PP1 binding to Ska1 is compromised, there is a 30% increase in phosphorylation of a MELT motif on Knl1 and a 20% increase in recruitment of Bub1 (*Figure 5B,C*). A similar (30–40%) increase in MELT phosphorylation and Bub1 levels also occurs upon mutation of Knl1 to inhibit its binding to PP1 (*Nijenhuis et al., 2014*). We sugggest that the pools of PP1 at the kinetochore are distinct and may play cooperative and specific roles at different stages of mitosis. It remains possible that there are unidentified targets whose dephosphorylation is more reliant on PP1 bound to Ska. The increase in Bub1 levels that we observe is also consistent with our earlier finding that Bub1 levels on kinetochores were increased in cells depleted of Ska3 (*Daum et al., 2009*). Although Ska does localize at kinetochores not attached to microtubules, it accumulates to its highest levels on kinetochores of cells at metaphase, the moment in time when quickly reversing checkpoint-dependent phosphorylation would be most useful in initiating anaphase.

The CTD of Ska1 has been structurally well characterized. It has the characteristics of a winged helix domain, a fold previously implicated in DNA binding and in mediating protein-protein interactions (*Abad et al., 2014*; *Schmidt et al., 2012*). It was shown that the Ska1 CTD binds microtubules through multiple sites and can bind both straight and curved microtubule protofilaments (*Abad et al., 2014*; *Schmidt et al., 2012*). The same Ska1 CTD is also required for PP1 binding. Because the Ska1 CTD binds to both microtubules and PP1, its functions in promoting chromosome alignment and anaphase onset might be a consequence of either or both activities. In previous work, it was shown that mutation of three conserved arginine residues in the Ska 1 CTD (R155A/R236A/R245A) compromised binding to microtubules in vitro and produced only a partial rescue of the metaphase delay caused by Ska1 depletion (*Abad et al., 2014*; *Schmidt et al., 2012*). We found that this mutant of Ska1 R3A was still able to co-precipitate PP1 in extracts from cells where both were expressed as transgenes (*Figure 4—figure supplement 3*). On the other hand, complete replacement of the CTD by a PP1-binding motif or by PP1 itself resulted in nearly complete rescue of all Ska1 depletion phenotypes, suggesting that microtubule binding by the Ska complex can be made dispensable. We previously reported that Ska depletion does not impair chromatid separation in cells induced to enter anaphase by application of a chemical Cdk1 inhibitor (*Sivakumar et al., 2014*). Based on that work and the data presented here, we propose that the microtubule-binding properties of Ska1 do not play a strong mechanical coupling function for kinetochore movement on microtubules. We were unable to find a Ska1 point mutant that retained microtubule binding but was deficient in PP1 binding to further test this idea. Given the clear molecular evidence for Ska binding to microtubules and to microtubule protofilaments, we favor the idea that microtubule binding to Ska serves a regulatory role. We propose that Ska1 CTD binds near the ends of kinetochore microtubules, where separated protofilaments are enriched. This binding in concert with Ska complex interactions with other kinetochore components, may control the local concentration, dynamics, or substrate specificity of the Ska-PP1 complex at kinetochores (*Figure 7F*).

PP1 was first identified as an important regulator that counters spindle checkpoint signaling in budding yeast and fission yeast (*Pinsky et al., 2009*; *Vanoosthuyse and Hardwick, 2009*). The N terminus of all Knl1 homologs from yeast to mammals contain conserved PP1-binding motifs (*Liu et al., 2010*). In budding and fission yeast, expression of Knl1 mutants unable to bind PP1 impaired the ability of the cells to overcome checkpoint signaling and strongly compromised cell growth (*Meadows et al., 2011*; *Rosenberg et al., 2011*). In *C. elegans* embryos, RNAi-mediated depletion of wild type Knl1 and its replacement with a PP1-binding mutant led to slow chromosome congression, delays at metaphase, and partial embryonic lethality (*Espeut et al., 2012*). The role of Knl1 binding of PP1 was studied in mammalian cells treated with nocodazole to disrupt microtubules and induce a strong spindle checkpoint arrest. Under these conditions, cells in which Knl1 was replaced with a PP1-binding mutant showed slower mitotic exit in comparison to controls when spindle checkpoint signaling was experimentally extinguished with a chemical inhibitor of Mps1 (*Nijenhuis et al., 2014*). Together these studies have led to a model in which PP1 binding by Knl1 is a key factor in opposing checkpoint signaling for promoting the onset of anaphase and mitotic exit.

However, one important result argues that this model does not fully explain the regulation of the metaphase-anaphase transition in normal mammalian cell mitosis. In mammalian cells with intact spindles, not treated with microtubule drugs, replacement of wild type Knl1 with a mutant Knl1 unable to bind PP1 results in only a modest, 10-min delay at metaphase (*Zhang et al., 2014*). In contrast, loss of PP1 recruitment by the Ska complex during normal mitosis causes a lengthy delay or complete arrest at metaphase. Importantly, Ska homologs have not been identified in budding or fission yeast, consistent with the importance of PP1 recruitment by Knl1 in those organisms. In *C. elegans*, a two-protein Ska complex is present (*Schmidt et al., 2012*). However, RNAi and mutant studies on Ska homologs in *C. elegans* embryos have not revealed an essential role in chromosome segregation (Arshad Desai, personal communication). Interestingly, an elegant approach for manipulating protein interactions within kinetochores in budding yeast at nanometer resolution indicated that recruitment of the yeast PP1 homolog to outer kinetochores was important for reversing Mps1 phosphorylations of Knl1 (*Aravamudhan et al., 2015*). In mammalian cells with intact mitotic spindles, our study suggests that Ska, an outer kinetochore protein complex, is a critical recruiter of PP1 in opposing spindle checkpoint kinase signaling at kinetochores.

Our data indicate that binding of Ska and binding of Knl1 to PP1 are independent, suggesting that multiple pools of kinetochore-associated PP1 may cooperatively counter kinase activities at kinetochores. Their functions may be additive, recruiting PP1 to the threshold level required for anaphase onset. Interestingly, similar to the Ska1 CTD, the N-terminal region of Knl1 adjacent to its PP1-binding motif also binds microtubules in vitro (*Cheeseman et al., 2006*). It is conceivable that the microtubule-binding domains of Knl1 and Ska1 may each regulate their associated PP1 pools, allowing them to be sensitive to the attachment status of the kinetochore. In addition, several other PP1-interacting proteins, including Cenp-E, SDS22 and Repo-man, have been identified as playing roles in mitosis (*Kim et al., 2010*; *Posch et al., 2010*; *Trinkle-Mulcahy et al., 2006*). However, these proteins, when expressed at endogenous levels, do not normally accumulate at kinetochores of metaphase chromosomes (*Kim et al., 2010*; *Eiteneuer et al., 2014*; *Wurzenberger et al., 2012*). During other stages of mitosis, prometaphase and anaphase, they may function in regulating PP1 activities on kinetochores, chromosome arms, and in the cytoplasm (*Eiteneuer et al., 2014*; *Wurzenberger et al., 2012*; *Qian et al., 2013*; *Qian et al., 2011*). In the future, it will be important to determine which specific protein phosphorylations are targeted by Ska-PP1 or by other PP1-binding proteins during mitosis. Finally, it is clear that PP2A, and possibly other phosphatases also play vital roles in regulating phosphorylation to control chromosome movement and cell cycle progression in mitosis (*Nijenhuis et al., 2014*; *Foley et al., 2011*; *Grallert et al., 2015*; *Kruse et al., 2013*; *Porter et al., 2013*; *Xu et al., 2014*; *Espert et al., 2014*).

In summary, here we make the surprising discovery that a chimeric Ska1-PP1 fusion lacking the microtubule-binding domain of Ska1 rescues nearly all the mitotic phenotypes observed upon Ska depletion, including delays in chromosome alignment and metaphase arrest. This rescue is fully dependent on the phosphatase activity of the chimera. Moreover, when expressed on its own, the phosphatase-dead Ska1-PP1 chimera has dominant phenotypes that closely mimic those of Ska depletion. Thus, rather than serving a mechanical coupling function between kinetochores and microtubules, the microtubule-binding properties of the Ska complex may primarily aid in coordinating PP1 recruitment to, or activity at, kinetochores. Our data suggest that PP1 recruitment is a critical function of the Ska complex for opposing mitotic kinases that destabilize kinetochore-microtubule attachment and that signal the spindle checkpoint. Thus, the Ska complex may integrate chromosome alignment at metaphase with full recruitment of PP1, thus opposing spindle checkpoint kinases signaling and promoting the metaphase-anaphase transition.

## Materials and methods

### Cell culture

All cell experiments were conducted with HeLa cells. Parental HeLa cells were obtained from ATCC, Manassas, VA. HeLA Tet-On cells were obtained from Clontech, Mountatin View, CA. HeLa cells stably expressing GFP-Histone H2B were provided by Geoff Wahl (*Kanda et al., 1998*). HeLa Ska1-GFP and HeLa Ska1ΔMTBD cell lines were obtained from Iain M Cheeseman (*Schmidt et al., 2012*). All lines were routinely tested and found to be free of mycoplasma but were not further authenticated.

All cell lines were grown in tissue culture dishes, culture flasks or chambered coverslips in 5% $CO_2$ at 37°C using complete DMEM media supplemented with 10% FBS, penicillin and streptomycin.

To synchronize HeLa cells, cultures were treated with 2 mM thymidine for 18–24 hr and released into media containing 3.3 µM nocodazole. Transient transfection for expression of plasmids was achieved using the Fugene 6 (Roche, Indianapolis, IN or Promega, Madison, WI or Mirus, Madison, WI) or Lipofectamine 3000 (Invitrogen, Carlsbad, CA) transfection reagent according to manufacturer's instructions. QuikChange Lightning site directed mutagenesis kit (Agilent technologies, Santa Clara, CA) was used to make point mutants of Ska1. Transfection of siRNA was done using Lipofectamine RNAi reagent (Invitrogen) according to manufacturer's instructions. SiGenome siRNA against Ska1, On target plus smartpool siRNA against Ska2 and Ska3 was obtained from DharmaconGE (Lafayette, CO) and these were used at 25–50 nM final concentration.

To generate the stable cell lines, HeLa Tet-On cells were transfected with pTRE2 vectors expressing siRNA resistant GFP-Ska1 or GFP-Ska1ΔCTD and selected with 300 µg/ml hygromycin (Clontech). Further screening of the clones to obtain stable cells was done in the presence of 150 µg/ml hygromycin.

## Live cell imaging

HeLa H2B-GFP or HeLa Tet-On cells were grown in Nunc chambered coverslips (Thermo Sci. Inc., Waltham, MA). In some instances, to visualize DNA in HeLa cells, a cell permeable Hoechst dye (33342; Invitrogen) was used at 25–50 ng/ml. Time-lapse fluorescence images were collected every 5 min for 24–48 hr using a Leica inverted microscope equipped with an environment chamber that controls temperature and $CO_2$, 40X objective, an Evolve 512 Delta EMCCD camera, and Metamorph software (MDS Analytical Technologies, Sunnyvale, CA). Time-lapse videos displaying the elapsed time between consecutive frames were assembled using Metamorph or ImageJ software. The first time frame denoting onset of nuclear envelope breakdown (NEB), metaphase chromosome alignment and anaphase onset/mitotic exit was recorded in Microsoft Excel and the interval from NEB to metaphase (alignment time), metaphase to anaphase (metaphase duration) or NEB to anaphase onset/mitotic exit was calculated. For every cell, mitotic duration was calculated and the data were depicted as scatter plots with mean and SEM. Only cells expressing the indicated constructs (determined by mCherry expression) were counted to determine mitotic duration. The indicated proteins were tagged either in N terminus or C terminus and in both cases similar results were obtained by live cell imaging. Further, to be certain that the mCherry tag was not specifically influencing mitotic progression, the proteins were also tagged with GFP and again similar results were obtained. For clarity, only images and results obtained with mCherry-tagged proteins are presented. In scatter plots, each dot represents one cell; long horizontal line depicts mean and whiskers denote SEM. Graphpad prism was used for statistical analysis.

## Immunofluorescence

HeLa cells were grown on glass coverslips or in Nunc chambered cover slides and treated as detailed in the figure legends. Cells were pre-extracted for 5 min using 1X PBS/PHEM solution containing 1% Triton X 100 supplemented with phosphatase inhibitors (Okadaic acid at 1 µM). Cells were then fixed in 2 or 4% paraformaldehyde/PHEM solution supplemented with phosphatase inhibitors for 15 min. Coverslips were washed in MBST, blocked in 20% Boiled Normal goat/donkey serum or 2% Bovine serum albumin (BSA) for 1 hr, and incubated overnight at 4°C or 1 hr at room temperature with primary antibodies. Samples were then incubated with secondary antibodies for 1 hr, stained with DNA dye, DAPI, and mounted using Vectashield (Vector Laboratories, Burlingame, CA). The following primary antibodies were used: ACA/CREST (Anti-Centromere antibodies from Antibody Inc, Davis, CA) and rabbit anti-Ska3 (*Daum et al., 2009*), sheep anti-PP1 (kind gift from Dr. Brautigan; used in *Figure 1A* images), goat anti-PP1 (Santa Cruz Biotechnology Inc.; used in *Figure 1C*), rabbit anti-pMELT (*Ji et al., 2015*) and rabbit anti-Bub1 (*Tang et al., 2001*). Secondary antibodies used were goat anti–rabbit or donkey anti-goat or goat anti-human antibodies conjugated to Cy3 or FITC (Jackson ImmunoResearch, West Grove, PA). The images were acquired using a Zeiss Axioplan II microscope equipped with a 100X objective (N.A. 1.4) or using 100X objective on Deltavision microscope (GE Healthcare, Pittsburgh, PA). Images were assembled using image J and Coreldraw (Corel Corp, Ottawa, Canada). Quantification of the immunofluorescence images was done as described

previously (*Daum et al., 2009*). The graphs depict mean fluorescence value with SEM obtained from at least 10 cells in each condition. Graphpad Prism (Graphpad, La Jolla, CA) was used to determine statistical significance among groups.

## Plasmids and protein purification

The Mis12-GFP, Ska1-GFP, Mis12Ska1-GFP plasmid construction has been described previously (*Sivakumar et al., 2014*). To construct fragments of Ska1, cDNA encoding indicated regions of Ska1 were PCR amplified from full length siRNA-resistant Ska1 and inserted in pCS2-GFP or GFP-N1 plasmid. pCS2-Myc-PP1 was made by inserting PP1γ into pCS2-Myc vector. Knl1$^{(34–81)}$ was made by amplying the region between 34 and 81 residues of full length Knl1 and inserting the cDNA in mCherry-N1 or pCS2-mCherry vector. Knl1$^{(34–81)4A}$ was made by mutating the RVSF motif in Knl1$^{(34–81)}$ to AAAA. Ska1ΔCTDPP1 or Ska1ΔCTDKnl1$^{(34–81)}$ or Ska1ΔCTDKnl1$^{(34–81)4A}$ fusion proteins were made by inserting PP1 or Knl1$^{(34–81)}$ or Knl1$^{(34–81)4A}$ after Ska1ΔCTD. All plasmids were verified by DNA sequencing.

Plasmid containing 6xHis-PP1α$^{7–330}$ was a gift from Wolfgang Peti (*Kelker et al., 2009*). Plasmids encoding GST-Ska3 and untagged Ska2/Ska1 were generous gifts from Iain M Cheeseman (*Schmidt et al., 2012*). To generate Ska1ΔCTD we introduced a single stop codons using site directed mutagenesis. We used a similar strategy to obtain GST-Ska3$^{1-343}$.

6xHis-PP1α 7–330 was expressed and purified essentially as previously described (*Kelker et al., 2009*) with following modifications. Prior to elution, Ni-NTA IMAC (Qiagen) was incubated with wash buffer containing additional 5 mM ATP and 5 mM MgCl$_2$ for 2 hr and subsequently washed with wash buffer. PP1 was further purified on Superdex 200 10/300 GL size-exclusion column in 50 mM HEPES pH 7.5, 150 mM KCl, 1 mM DTT.

Ska1ΔCTD/Ska2/Ska3$^{1-343}$ and Ska1$^{FL}$/Ska2/Ska3$^{1-343}$ for gel filtration experiments were purified according to previously established protocol for Ska complex purification (*Schmidt et al., 2012*) with following modification. Immediately after the cleavage of the GST tag, proteins were concentrated using Amicon Ultra centrifugal filters and subsequently purified on Superdex 200 10/300 GL size-exclusion column (GE Healthcare/Life Sciences) in 50 mM Tris pH 7.5, 150 mM KCl, 1 mM DTT.

To generate complexes 100 nM 6xHis-PP1α 7–330 was incubated on ice for 1 hr with Ska1ΔCTD/Ska2/Ska3$^{1-343}$ or Ska1$^{FL}$/Ska2/Ska3$^{1-343}$ in 1:2 molar ratio before loading on Superose 12 PC 3.2/30 size-exclusion column (in 50 mM HEPES, 150 mM KCl, 2 mM MnCl$_2$ and 1 mM DTT) and 50 μl fractions were collected. Immunoblots were performed using antibodies raised against Ska3 (*Daum et al., 2009*) and hPP1 (gracious gift from David Brautigan). The quantification of total PP1 was determined as a sum of the densitometry measurements (ImageJ) of all visualized fractions.

To express the Ska proteins in bacteria for in vitro binding experiments, pGEX-Duet encoding GST-Ska1 and untagged Ska2 or PGEX-Ska3 were individually transformed into BL21 (DE3) pLysS cells (Invitrogen) and protein expression was induced using 0.1 mM IPTG at 16–18°C overnight. Bacterial cell pellets were lysed with lysis buffer (25 mM Tris-HCl, pH 8.0, 150 mM NaCl, 10% (v/v) glycerol, 1–5 mM DTT, 0.1% Triton) and sonicated. The lysate was then cleared by centrifugation at ~25,000 g for 1 hr. Supernatant was incubated over glutathione sepharose 4B beads (GE Healthcare) for 1–2 hr at 4°C. Beads were washed with lysis buffer 3–4 times. The GST-Ska proteins were eluted using glutathione, further purified and concentrated using PD-10 desalting columns (GE Healthcare) and Amicon Ultra centrifugal filters (EMD Millipore, Billerica, MA).

## In vitro binding assays and microscale thermophoresis (MST)

Bacterially purified GST-Ska proteins were bound to glutathione sepharose 4B beads (GE Healthcare). S$^{35}$-labeled Myc-PP1 was obtained by in vitro translation using rabbit reticulocyte lysates (Promega). The Myc-PP1 was incubated with GST-Ska protein bound beads for 1 hr at room temperature. Beads were washed using wash buffer (50 mM Tris HCl, pH 7.5, 100 mM KCl, 0.05% Triton) 3–4 times and 2X SDS sample buffer was added. The proteins were resolved using SDS-PAGE gel, and the S$^{35}$ signal was analyzed using a phosphor-imager (Fuji, Burlington, NJ).

To perform MST analysis, the GST-Ska1ΔCTD and His-PP1 were purified from bacteria. GST-Ska1ΔCTD was purified as described above and His-PP1γ was purified as described previously (*Peti et al., 2013*). Briefly pET28a-His-PP1γ and a chaperone pGro7 (expressing GroES-GroEL) was transformed into BL21 cells and protein expression was induced overnight at 10°C using L-Arabinose

and IPTG. The bacterial pellets were lysed in lysis buffer (50 mM Tris pH8.0, 5 mM imidazole, 700 mM NaCl, 1 mM MnCl2, 0.1% Triton X-100) and sonicated. The lysate was cleared by centrifugation at ~25,000 g for 1 hr. The supernatant was incubated over Ni$^{2+}$-NTA resin (Qiagen, Valencia, CA) to allow His-PP1γ binding. For MST analysis, the His tag was cleaved using Thrombin. The GST tag was cleaved from Ska1ΔCTD using 3C protease. The untagged proteins were eluted, further purified using Superdex size exclusion columns and concentrated using Amicon ultra centrifugal filters (GE Healthcare). Finally both Ska1ΔCTD and PP1 proteins were exchanged into MST buffer (25 mM HEPES, 50 mM NaCl, 1 mM TCEP).

For MST, PP1 was covalently coupled to a fluorophore by incubating 200 µl of PP1 at a concentration of 40 µM with 1 µl of 40 mM Alexa-Fluor 488-N-hydroxysuccinimide ester (Molecular Probes/Life Technologies, Grand Island, NY) dissolved in 100% dimethyl sulfoxide) for 30 min in the dark at room temperature. The labeled protein was separated from free dye by applying the mixture to a G25 column that had been equilibrated with 9 ml of protein storage buffer. Serial dilutions (1:1) of Ska1 CTD were made in 15 successive 10 µl reactions, resulting in 16 samples, with the highest concentration of Ska1 CTD being 40 µM. Each of the reactions was mixed with 10 µl 800 nM labeled PP1. Thus, the final concentration of the labeled protein was 400 nM in all samples, and the final highest concentration of Ska1 CTD was 20 µM; all reaction mixtures were supplemented with Tween-20 (NanoTemper, LLC, Munich, Germany) to a final concentration of 0.05% (v/v). After incubation at room temperature for approximately 30 min., all the reactions were loaded into standard treated capillary tubes, and the final measurements were taken in a Monolith NT.115 instrument (Nanotemper LLC, Munich, Germany). The instrument's LED (illumination) power was set to 25% and the MST laser power was set to 40%. Measurements were performed at ambient temperature, ca. 23°C. The times of data acquisition were 5 s before the activation of the MST laser, 30 s with the laser on, and 5 s after extinguishing the laser. Data analysis was performed in PALMIST (biophysics. swmed.edu/MBR/software.html) using the T-Jump mode (*Scheuermann et al., 2016*). A negative control experiment in which Soybean Trypsin Inhibitor (Worthington Biochemical Corp., Lakewood, NJ) was titrated into labeled PP1 under identical conditions showed only a weak trend in T-jump behavior (data not shown).

## Immunoprecipitation, Western blotting and blot quantification

Antibodies (GFP, Ska3 or Knl1- all produced in-house) were coupled to Affi-prep Protein A beads (Biorad) at a concentration of 1 mg/ml. Whole cell HeLa cell extracts were lysed in buffer (25 mM Tris-HCl pH 7.5, 75 mM NaCl, 5 mM MgCl$_2$, 0.1% NP-40, 0.4% Triton X 100, 1–5 mM DTT, 0.5 µM okadaic acid, 5 mM NaF, 0.3 mM Na$_3$VO$_4$, 10 mM β Glycerophosphate, 50 units/ml Turbo-nuclease (Accelagen, San Diego, CA))containing protease inhibitor cocktail (Roche). Lysate was cleared by high-speed centrifugation at 20,817 g for 20 min. The supernatant after centrifugation was added to antibody coupled beads and incubated at 4°C for 2–4 hr by end-over-end rotation. After incubation, the beads were washed three to four times and the immunoprecipitation reaction was terminated by addition of 2X SDS sample buffer. The immuno-precipitated proteins were resolved by SDS-PAGE electrophoresis and blotted with specific antibodies. Primary antibodies included rabbit anti-GFP antibody (made in house), mouse anti-Myc (Roche), rabbit anti-Knl1 (made in house) and rabbit anti-Ska3 (made in house). Membranes were washed in TBS/0.05% Tween 20 (TBST), and then incubated with secondary antibodies in 5% NFDM/TBST. Secondary antibodies were Dylight fluorescent dye conjugated (Cell Signaling, Danvers, MA), and the membranes were analyzed with the Odyssey CLx Infrared Imaging system (LICOR, Lincoln, NE). The protein bands on the membranes were quantified using Image Studio software (LICOR). Every experiment was repeated at least three times, and the quantification shows the mean with SEM in each case.

## Acknowledgements

We thank Iain M Cheeseman for plasmids and HeLa cell lines exparessing GFP-Ska1 and HeLa GFP-Ska1ΔCTD. We thank Geoff Wahl for the HeLa cell line expressing GFP-Histone H2B. We thank Wolfgang Peti for PP1 plasmid. We thank David L Brautigan for PP1 antibody, plasmids, and for useful advice. We also thank members of the Yu, Gorbsky, and Stukenberg laboratories for help with experiments and useful suggestions. We would especially like to thank Dr. Haishan Gao in Dr. Yu's laboratory for his help with protein purification. This study was supported by funding from National

Institutes of Health, grant number RO1GM111731 to GJG and grant number R01GM081576 to PTS, from the McCasland Foundation to GJG, and from the Clayton foundation to HY.

## Additional information

### Funding

| Funder | Grant reference number | Author |
|---|---|---|
| National Institute of General Medical Sciences | RO1GM111731 | Gary J Gorbsky |
| Clayton Foundation | | Hongtao Yu |
| McCasland Foundation | | Gary J Gorbsky |
| National Institute of General Medical Sciences | R01GM081576 | P Todd Stukenberg |

The funders had no role in study design, data collection and interpretation, or the decision to submit the work for publication.

### Author contributions

SS, Conception and design, Acquisition of data, Analysis and interpretation of data, Drafting or revising the article; PŁJ, Performed and interpreted the gel filtration assays, Acquisition of data, Analysis and interpretation of data; QQ, Assisted in some of the protein purifications and carried out some supplementary experiments, Acquisition of data, Contributed unpublished essential data or reagents; CAB, Helped to design and carry out the MST experiments, Acquisition of data, Analysis and interpretation of data, Drafting or revising the article; PTS, HY, GJG, Conception and design, Analysis and interpretation of data, Drafting or revising the article

### Author ORCIDs

Gary J Gorbsky, http://orcid.org/0000-0003-3076-4725

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
