## [Decision Letter]

Thank you for submitting your work entitled "The Ska complex drives the metaphase-anaphase cell cycle transition by recruiting protein phosphatase 1 to kinetochores" for consideration by *eLife*. Your article has been favorably evaluated by Jessica Tyler (Senior editor) and three reviewers, one of whom, Sue Biggins, is a member of our Board of Reviewing Editors.

The reviewers have discussed the reviews with one another and the Reviewing Editor has drafted this decision to help you prepare a revised submission.

Summary:

This manuscript identifies the Ska1 kinetochore protein as a potentially new PP1 regulatory subunit at the human kinetochore. PP1 activity is essential to reverse the spindle checkpoint and promote anaphase, so the findings in the paper are significant to the field of mitosis as well as to those interested in phosphatase regulation. The authors suggest that PP1 and Ska1 interact through the C-terminal domain (CTD) of Ska1 and that the phenotypes of cells containing mutants in this domain are due to defects in PP1 recruitment to kinetochores. While the findings in the manuscript are potentially exciting, the major concern with the work is that a direct interaction between PP1 and the Ska1 CTD was not demonstrated. This leads to the concern that the association could be indirect, so the following work needs to be completed for a revision to be accepted.

Essential revisions:

1) Direct binding between PP1 and Ska1 CTD must be demonstrated. The authors should make recombinant proteins, demonstrate PP1 activity to ensure it is functional, and perform ITC or SEC (in this case, with chromotograms reporting A280 or A215 if the CTD has no Trp residues of each protein alone, the complex and molecular weight standards in addition to gels; ITC is preferable) to show an interaction.

2) Given the difficulty of identifying a specific PP1 binding mutant in Ska1, we would like to see that the addition of a short peptide with an RVXF motif on the Ska1 mutant will rescue. This will allow dynamic recruitment of PP1 and be more meaningful than the constitutive PP1 fusion.

---

## [Author Response]

We thank the reviewers for critical evaluation of our manuscript and for the useful suggestions below. We agree that PP1 plays an essential role to reverse the spindle checkpoint in mitosis. We are happy that the reviewers find our findings “significant to the field of mitosis”. We have taken the reviewers’ suggestions to heart, and performed additional experiments to establish direct binding between PP1 and Ska1 CTD. We present our results below and have included them in the manuscript. With the new data, our original conclusions are further strengthened. We hope that the reviewers will support acceptance of this manuscript at *eLife*. Please find a detailed response to reviewers’ comments below.

*1) Direct binding between PP1 and Ska1 CTD must be demonstrated. The authors should make recombinant proteins, demonstrate PP1 activity to ensure it is functional, and perform ITC or SEC (in this case, with chromotograms reporting A280 or A215 if the CTD has no Trp residues of each protein alone, the complex and molecular weight standards in addition to gels; ITC is preferable) to show an interaction.* We purified PP1 and Ska1 CTD from bacteria (*E. coli*) and performed Microscale Thermophoresis (MST) assay. Active PP1 is difficult to purify in large quantity, which is a requirement for ITC. MST is a relatively new and sensitive technique to measure protein interactions in vitro. To perform MST, we sought help from an expert, Chad A Brautigam, who has been added as a co-author. We determined that the K_d_ of the Ska1 CTD-PP1 interaction is 1.5 μM. This is comparable to other protein-protein interactions that allow dynamics. We have presented these data in Figure 3 of the revised manuscript. Note that these data are also supported by the gel filtration (SEC) data that we originally included that shows a direct interaction between PP1 and the Ska complex with recombinant proteins. These SEC data demonstrated a requirement for Ska1 CTD in direct PP1 binding. We could not, however, detect an interaction between PP1 and the isolated Ska1 CTD using SEC, presumably due to the moderate affinity of this interaction. Concentration of the Ska complex at kinetochores may allow cooperative binding, or other components of the Ska complex may contribute to PP1 binding.

2) Given the difficulty of identifying a specific PP1 binding mutant in Ska1, we would like to see that the addition of a short peptide with an RVXF motif on the Ska1 mutant will rescue. This will allow dynamic recruitment of PP1 and be more meaningful than the constitutive PP1 fusion.

We thank the reviewers for this wonderful suggestion. As suggested, we fused the fragment of Knl1 (from amino acids 34 to 81 to human Knl1) that contains the PP1-binding motif (but importantly lacks the Knl1 microtubule-binding domain) to Ska1ΔCTD and performed live cell imaging. As the control, we used the same fragment of Knl1 where the PP1-binding RVSF motif was mutated to AAAA. We found that the Ska1ΔCTD fused to the wild type Knl1 fragmentrescued mitotic defects caused by Ska1 depletion while Ska1ΔCTD fused to Knl1 AAAA did not. We have presented these data in Figure 6 and in the second paragraph of the subsection “Ska-mediated kinetochore recruitment of PP1 promotes anaphase onset”.